# Antibacterial Activity of Silver Nanoflake (SNF)-Blended Polysulfone Ultrafiltration Membrane

**DOI:** 10.3390/polym14173600

**Published:** 2022-08-31

**Authors:** Gunawan Setia Prihandana, Tutik Sriani, Aisyah Dewi Muthi’ah, Siti Nurmaya Musa, Mohd Fadzil Jamaludin, Muslim Mahardika

**Affiliations:** 1Department of Industrial Engineering, Faculty of Advanced Technology and Multidiscipline, Universitas Airlangga, Jl. Dr. Ir. H. Soekarno, Surabaya 60115, Indonesia; 2Department of Research and Development, PT. Global Meditek Utama, Sardonoharjo, Ngaglik, Sleman, Yogyakarta 55581, Indonesia; 3Centre of Advanced Manufacturing & Material Processing (AMMP Centre), Department of Mechanical Engineering, Faculty of Engineering, Universiti Malaya, Kuala Lumpur 50603, Malaysia; 4Department of Mechanical and Industrial Engineering, Faculty of Engineering, Universitas Gadjah Mada, Jalan Grafika No. 2, Yogyakarta 55281, Indonesia

**Keywords:** polysulfone membrane, silver nanoflake, good health, antibacterial activity, *Escherichia coli*, water treatment

## Abstract

The aim of this research was to study the possibility of using silver nanoflakes (SNFs) as an antibacterial agent in polysulfone (PSF) membranes. SNFs at different concentrations (0.1, 0.2, 0.3 and 0.4 wt.%) were added to a PSF membrane dope solution. To investigate the effect of SNFs on membrane performance and properties, the water contact angle, protein separation, average pore size and molecular weight cutoffs were measured, and water flux and antibacterial tests were conducted. The antimicrobial activities of the SNFs were investigated using *Escherichia coli* taken from river water. The results showed that PSF membranes blended with 0.1 wt.% SNFs have contact angles of 55°, which is less than that of the pristine PSF membrane (81°), exhibiting the highest pure water flux. Molecular weight cutoff values of the blended membranes indicated that the presence of SNFs does not lead to enlargement of the membrane pore size. The rejection of protein (egg albumin) was improved with the addition of 0.1 wt.% SNFs. The SNFs showed antimicrobial activity against *Escherichia coli*, where the killing rate was dependent on the SNF concentration in the membranes. The identified bacterial colonies that appeared on the membranes decreased with increasing SNF concentration. PSF membranes blended with SNF, to a great degree, possess quality performance across several indicators, showing great potential to be employed as water filtration membranes.

## 1. Introduction

Clean water scarcity has become one of the greatest global challenges that need to be solved [1,2,3]. Different water separation technologies have been developed to tackle the problem [4]. For this purpose, economical and efficient water separation technologies are needed. Among other water separation technologies, such as flotation, skimming, ultrasonic treatment and coagulation [5,6,7,8,9], membrane separation has attracted more attention due to its performance, high efficiency, low-cost fabrication and operation [10,11,12]. To date, ceramic and polymeric membranes have been developed and used in water separation technologies [13,14]. Compared to ceramic membranes, polymeric membranes display outstanding performance, such as controllable membrane microstructures, good flexibility and easy fabrication [4,15].

Filtration utilizing membrane-based polymer technology such as microfiltration (MF), ultrafiltration (UF), nanofiltration (NF) and reverse osmosis (RO) is a potent solution to overcoming the scarcity of the clean and safe water supply [16]. The outstanding combination of water flux and rejection have made polymeric membranes widely used in various water filtration applications [17,18]. In addition, polymeric membranes possess a microporous, open spongy morphology of tiny pores ranging from 0.03 µm to 10 µm in diameter [19,20], which is suitable for all water filtration purposes.

Different types of commercial membranes have been used and developed using common polymers, such as poly (vinylidene fluoride) (PVDF) [14,21], polyether sulfone (PES) [22,23], poly (acrylonitrile) (PAN) [24,25], polyimide (PI) [26,27], polyurethane (PU) [28,29], poly (propylene) (PP) [30,31] and poly (sulfone) (PSF) [32,33].

Among those polymeric membranes, PSF is an ideal choice for membrane development due to its availability, chemical stability and high mechanical strength and can be used in a wide range of temperature and pH levels [34,35]. PSF has a tensile strength between 78 and 83 MPa; hence, it exhibits excellent mechanical properties that favor its usage as a membrane in wastewater treatment [36]. However, hydrophobicity is considered to be a disadvantage of PSF membranes, as it contributes to membrane fouling [37,38]. To further improve the properties and filtration performance of PSF membranes, many modification methods, such as polymer blending [39,40,41], incorporating nanoparticles [42,43], surface chemical modification [42,44], interfacial polymerization [45,46] and the incorporation of additives, have been conducted. It is known that the additives modulate the hydrophilicity of the surface of PSF membrane and control its morphology and pore formation [47,48,49].

Recently, various types of additives have been used to modify the PSF membrane and to attain material and filtration improvements in terms of water flux and solute rejection. Kusworo et al. prepared PSF membranes by adding TiO_2_ nanoparticles to the dope solution. The study revealed that the introduction of TiO_2_ nanofillers enlarged the finger-like structure of the membrane, yielded a decreased contact angle and increased the pure water flux [50].

Graphene oxide (GO) has also been evaluated as a nanofiller in PSF ultrafiltration membranes. Differently shaped GO materials, flat and crumpled GO, were used in the membrane fabrication via the phase inversion method. The experimental result shows that the shape of the GO material affects the dispersion and viscosity of the dope solution and results in changes to the membrane structures (surface hydrophilicity and porosity) and performance (permeability and rejection) [51]. Alosaimi explored the effect of the incorporation of oxidized carbon nanotubes (CNTOxi) into PSF membranes. The addition of a hybrid nanocomposite improves the thermal stability, mechanical properties and adsorption capacities of the membranes due to the good dispersion of organoclay and carbon nanomaterials [52].

Cellulose nanocrystals were used as an additive to PSF to form nanocomposite mixed-matrix membranes (NMMMs). It was reported that fabricated NMMMs have triple the pure water flux and an improvement in the methylene blue (MB) rejection efficiency [53]. MOFs have also been prepared as composite membranes. A high pure water flux membrane was obtained by incorporating metal-organic framework (MOF) MIL-100 (Fe) nanoparticles into a PSF composite membrane. In addition, fabricated PSF membranes have a 99% rejection rate of MB and excellent anti-fouling performance [54].

In water treatment, bacterial and viral penetration is the obstacle to avoid. In order to overcome this problem, silver nanoparticles (SNs) incorporated into PSF membranes have been shown to exhibit antimicrobial properties. Silver is known for its good electrical and thermal conductivity, chemical stability and antibacterial activity [55]. Its antibacterial property is due to the availability of Ag^+^, which prolongs the prevention of bacterial adhesion. In addition to its antibacterial property, as an additive, SNs improved membrane hydrophilicity and reduced membrane fouling [56]. Moreover, membranes with incorporated nanosilver showed different porosities as a function of additive particle size [57]. Pal et al. [58] reported that the toxicity of SNs to bacteria is affected by SN size and shape, since in terms of the active facet, different shapes may have different effective surface areas. Nanoflakes can be defined as nanoscale particles which have three dimensions at the nanoscale [59], as shown in Figure 1. On the other hand, bacteria are susceptible to mechanical forces [60,61], and the sharp edges of nanoflakes could be beneficial in penetrating the bacterial cell membrane. In the literature, there are several works related to silver nanoflakes (SNFs); however, there is no scientific study about the characteristic properties of SNFs incorporated in PSF membranes, particularly on the effect of SNFs on the average pore size of the membrane, the separation of molecular weight protein solution (egg albumin) and the ultrafiltration performance of the fabricated membrane.

This work focuses on preparing bare and composite PSF membranes. The membranes were fabricated by introducing different quantities of SNFs (0, 0.1, 0.2, 0.3 and 0.4 wt.%) to the PSF (22 wt.%) membrane solution. Contact angle and adhesion work were the techniques used to characterize the membranes. Membrane performance was investigated by using a water flux test, a protein rejection test, an average pore size assessment and a molecular weight cutoff measurement. Finally, the antibacterial property was evaluated by quantifying the number of *Escherichia coli* attached to the membrane surface.

## 2. Materials and Methods

### 2.1. Materials

The PSF dope solution was prepared using Udel P-3500 (Solvay SA, Brussels, Belgium). Egg albumin (EA) (45 kDa) was obtained from HiMedia Laboratories Pvt Ltd., Mumbai, Maharashtra 400086, India. Chromocult^®^ coliform agar and N-Methyl-2-pyrrolidone (NMP), used as a solvent, were obtained from Merck KGaA, Darmstadt, Germany. Silver nanoflake powder (thickness of 80–500 nm), as shown in Figure 1, was obtained from Nanostructured & Amorphous Materials, Inc. Katy, TX 77494, USA. Pure water was used for the membrane’s fabrication and water flux test.

### 2.2. Membrane Fabrication

The PSF membrane was prepared using the phase inversion method. PSF, as a polymeric binder, was added to the solvent (NMP) until it was completely dissolved. Once a homogeneous solution was formed, SNF powder was slowly added to the dope solution at different concentrations (0, 0.1, 0.2, 0.3 and 0.4 wt.%), as presented in Table 1, and stirred with a hot plate magnetic stirrer. The dope solution of PSF/SNF was poured onto a glass plate and casted using a film applicator (Elcometer, Manchester M43 6BU, UK) at a thickness of 200 µm. The solution formed on the glass plate was gently transferred to a gelatinization bath. The bath was composed of pure water (non-solvent), and during the gelatinization process, an instantaneous liquid–liquid demixing process controlled the membrane surface and matrix phase inversion process, resulting in a membrane with a finger-like structure and a dense skin on its surface, as illustrated in Figure 2 [62].

### 2.3. Membrane Characterization

Pure water at a fixed volume was deposited onto the surface of the dry membrane. The image of the deposited water was captured using an AM73915MZTL Dinolite Edge 3.0 digital microscope. The angle of the water drop was calculated using AutoCAD software. To guarantee a valid result, the contact angle for each membrane was measured three times and the values were acquired by averaging the three repetitions. From the contact angle values obtained, the work of adhesion $\left(\omega_{A}\right)$, i.e., the surface energy required to drag water from a membrane surface, can be determined as follows [63]:(1)$$\omega_{A}=\gamma_{B}(1+\cos\theta )\;$$
where $\gamma_{B}$ is the water surface tension and θ is the contact angle.

### 2.4. Equilibrium Water Content

The fabricated PSF membranes were cut to the required size, immersed in pure water for 24 h and weighed directly after wiping the free water from the membrane surface. The membranes were then dried out and weighed again. The water content of the membrane was calculated by the following [64]:(2)$$\%\mathrm{WC}=\frac{W_{w}-W_{d}}{W_{w}}\times 100\;$$
where $W_{w}$ and $W_{d}$ are the weights of the wet and dry membranes, respectively. The measured data were then analyzed using one-way ANOVA with SPSS Software version 16, SPSS Inc., Chicago, IL, USA, and the level of significance was set at 0.05.

### 2.5. Filtration Experiments

#### 2.5.1. Water Flux Test

Filtration experiments were conducted in a stirred dead-end cell (Sterlitech UHP-62, Sterlitech Corp. Kent, WA, USA), as shown in Figure 3, with a diameter of 62 mm and an effective membrane area of 30 cm^2^. Nitrogen gas at a pressure of 2 bar was delivered into the dead-end cell to provide pressure to the tested water. Data acquisition was used to record the weight of the permeate water passing through the membrane pores. The following formulas were used to calculate the volumetric flux and permeability [65]:(3)$$Flux\;\left(J_{v}\right)=\frac{Q}{A\Delta t}$$
(4)$$Permeability\;\left(L_{p}\right)=\frac{J_{v}}{\Delta P}$$
where Q is the quantity of the permeate water (in L) during the sampling time, Δt (in h) is the time difference, A is the area of the membrane (in m^2^) and ΔP is the pressure difference (in bar).

#### 2.5.2. Protein Separation

EA solution at a concentration of 0.1 wt.% was prepared in phosphate-buffered solution (pH = 7.2). The protein rejection experiment was conducted at a fixed pressure of 2 bar using the dead-end cell test. An N4S UV-visible spectrophotometer was used to measure the concentration of the protein at a wavelength of 280 nm. Proteins contain amino acids (tryptophan) that absorb light in the UV spectrum at wavelength of 280 nm [66]. The solute rejection (SR) was determined with the following [67]:(5)$$\%\mathrm{SR}=\left[1-\frac{C_{p}}{C_{f}}\right]\times 100\;$$
where $C_{p}$ and $C_{f}$ are the protein concentrations in the permeate and feed solution, respectively.

#### 2.5.3. Measurement of Average Pore Size

The results from the ultrafiltration test were used to calculate the average pore size of the membrane surface. To do so, the molecular weight of solutes with a solute rejection (SR) of more than 80% was used in the equation below [68]:$$\bar{R}=100\left[\frac{\propto }{\%\mathrm{SR}}\right]$$
where $\bar{R}$ is the average radius of the pore size and ∝ is the solute radius, represented by the Stokes radius acquired from the solute molecular weight.

### 2.6. Molecular Weight Cutoff (MWCO)

There is a linear correlation between MWCO and the membrane’s pore size. In every case, the membrane MWCO is investigated by identifying an inert solute of the smallest molecular weight that shows a protein rejection of 80–100% in an ultrafiltration test. In this study, EA was selected as a protein due to its percentage rejection of the PES/SNF-blended membrane [69].

### 2.7. Antibacterial Experiment

The PSF membranes were evaluated for the Gram-negative bacteria *Escherichia coli* in the untreated river water. Feng et al. [70] and Dadari et al. [71] found that heavy metals such as silver have been verified to be effective in antibacterial performance against Gram-negative *Escherichia coli* and Gram-positive *Staphylococcus aureus*. This is because the presence of silver in bacterial cells may start protein deposition in cells and may deactivate them. Either Gram-negative or Gram-positive experiments yield identical outcomes. Therefore, coliform agar was selected to grow *Escherichia coli* instead of *Staphylococcus aureus.* Untreated water, such as irrigation water or river water in populated areas—with some people having limited access to water supply still needing to fulfill their daily water intake—may have been contaminated by bacteria and viruses [72,73].

Prior to the test, coliform agar was prepared by autoclaving the liquid media (25.6 g of coliform agar in 1 L of pure water) and cooling it down in the Petri dish. During the antibacterial experiment, the fabricated membranes were used to filter out bacteria in the contaminated water (river water). The water sample was taken from the Trasi River in Sleman, Yogyakarta province, Indonesia, which is used for irrigation purposes. The tested membranes were then placed on the surface of the prepared coliform agar. The EMB agar plates were then incubated at 35 °C for 24 h. Table 2 presents the properties of the river water used in this study.

The number of bacteria present on the membrane surface was analyzed with ImageJ software. For the ImageJ analysis, the images of the membrane surface were converted to 8-bit mode. Brightness and contrast were then adjusted to eliminate noise in the image background. Automatic thresholding was then utilized to convert images to binary (black and white), where the black areas represent the bacterial colonies and white areas represent the surface of the membranes [74,75].

## 3. Results and Discussion

### 3.1. Contact Angle Analysis

To examine the effect of SNF on surface hydrophilicity, the contact angles of the fabricated membranes were measured. Figure 3 presents the water contact angle of the PSF membranes at different concentrations of SNFs. As shown in Figure 3, bare PSF membranes (0 wt.% SNFs) have the highest contact angle, whilst the most hydrophilic surface was achieved at an SNF concentration of 0.1 wt.%. The addition of SNFs to polysulfone membranes caused a reduction in the water contact angle of the modified membranes. This is in accordance with the study of Kasraei et al. [76]. The contact angle of SNF-PSF membranes slightly increased with the increase in SNF concentration. This result indicates that the introduction of SNFs could improve the hydrophilicity of the membrane in comparison to the bare PSF membrane. However, the membrane hydrophobicity increases when the SNF concentration is amplified from 0.1 to 0.4 wt.%. This is possibly due to the creation of stacked nanoflake particles at SNF concentrations that exceed 0.1 wt.%, forming clusters of SNFs on the membrane surface, thus leading to an increase in the water contact angle. Subsequently, the contact angle values were used to calculate the work of adhesion, as presented in Figure 4. The highest value of the work of adhesion was obtained for the PSF membrane with 0.1 wt.% SNFs. These results indicate that at a certain concentration, the addition of SNFs improves the surface hydrophilicity of the membrane to its maximum value.

Contradictory to the result in this study, Bouchareb et al. [77] reported that the membrane contact angle decreases with increasing silver nanoparticles-graphene oxide (GO) concentration. They combined silver nanoparticles with graphene oxide as the additive in their study. The residual hydrophilic oxygen-based functional groups of graphene oxide and silver nanoparticles were attributed to the decrease in the superficial retention of the membrane.

### 3.2. Equilibrium Water Content Study

Figure 5 shows the equilibrium water content of the membranes with different concentrations of SNFs. The bare PSF membrane has a maximum water content of 65.4±0.9%. The water content of the membrane slightly increases with the increase in SNF concentration. At an SNF concentration of 0.1 wt.%, the membrane achieved the highest water content (67.2±0.6%), as presented in Table 3, and became the most hydrophilic membrane compared to the others. When the membrane hydrophilicity increases, the membrane is capable of transporting more water through, hence increasing the water content of the membrane [67,78].

A one-way ANOVA test was conducted to determine if there is a significant difference in the equilibrium water content between at least two groups. The results showed that p=0.023, below the significance level of 0.05, as presented in Table 4. Therefore, we reject H_0_ and conclude that there was a statistically significant difference in equilibrium water content between at least two groups.

### 3.3. Pure Water Flux Test Experiments

Figure 6 shows the pure water flux of the fabricated membranes measured at different concentrations of SNFs. As shown in Figure 5, pure water flux was increased from 88 to 116 L m^−2^ h^−1^ bar^−1^ by adding 0.1 wt.% SNFs, which is the highest amongst all the tested membranes. This is attributed to the improvement in membrane hydrophilicity, which is one of the essential factors in the enhancement of pure water flux. This result is consistent with Bilici et al. [79], who stated that the increase in pure water flux was probably due to the increase in hydrophilicity. However, at higher SNF concentrations, the pure water flux started to decline. At higher concentrations, as more SNFs cover and block the membrane pores since extra SNFs are introduced, more SNFs that are present in the membrane tend to cause extra blockage of the pores, compliant with what has been reported by Mollahosseini et al. [80] where pore blockage occurred due to higher SNF content.

### 3.4. Protein Separation

Figure 7 shows the egg albumin (EA) rejection of the fabricated membranes, whose numerical rejection values are summarized in Table 3. As presented in Figure 6, the PSF membrane at 0.1 wt.% SNFs has a slightly higher EA rejection compared to the bare PSF membrane (0 wt.% SNFs). However, EA rejection of the PSF/SNF blended membrane slightly decreases at higher concentrations of SNFs. It can be observed that the EA rejection is not significantly changed at various concentrations of SNFs, since the addition of SNFs is less than 0.5 wt.% and the percentage increment is only 3.4%.

Contrary to the results of the present study, Mollahosseini et al. [80] reported that less protein macromolecules could pass through the membranes when the silver nanoparticle concentration in the membrane increased. They utilized higher concentrations of silver nanoparticles in their research, which included 0, 0.5, 2 and 4 wt.%. It is possible that the addition of more than 0.5 wt.% silver nanoparticles into the polysulfone dope solution caused pore reduction, further resulting in higher protein rejection.

### 3.5. Measurement of Average Pore Size

Figure 8 presents a summary of the average pore size of the membranes. Based on the EA rejection data, all membranes have a protein rejection value between 86 and 89%, where the PSF membrane with 0.1 wt.% SNFs has the highest rejection (89.9%). These results can be further accounted for given that those membranes have an average pore size in the range of 36 Å–38 Å. There are no significant changes to the membranes’ average pore size at SNF concentrations between 0.1 and 0.4 wt.%. This is due to the relatively low concentration of SNFs in the dope solution, as supported by the membrane surface morphology presented in Figure 9.

In contrast to the result in this study, Li et al. [81] found that the average pore size of the membranes increased due to the increase in the total solid content (1–5 wt.%). This is possibly due to the amount of additives added to the membrane casting solution, where, to suppress defects, the small amount induced a denser skin layer, whereas the large amount of particles added led to a looser skin layer.

### 3.6. Molecular Weight Cutoff Measurement

In this study, EA with a molecular weight of 45 kDa was used to investigate the membrane MWCO. Based on the results obtained, the membrane MWCO is not affected by the introduction of SNFs at concentrations of 0.1–0.4 wt.% (Table 3). The size of the nanoflakes was still within the range of the permitted concentrations used in the membrane dope solution; therefore, the MWCO value remains the same.

Coincidently, similar results have previously been reported by Mahmoudi et al. [80]. Their research claimed that the addition of silver nanoparticles–graphene oxide nanoplates has no significant effect on the protein rejection of the membranes due to the low concentration of the additives (0.1–0.5 wt.%). Furthermore, the results indicated acceptable rejection values for the membranes.

### 3.7. Evaluation of the Antimicrobial Activity

Figure 10 presents the results of the antibacterial test for the fabricated membranes at different concentrations of SNFs. The images of the membrane area contaminated by bacterial colonies were taken and quantitatively measured using the image processing method. The images were then binarized and adjusted to detect the colonies’ growth. Both the obtained images and the binarized images were compared to ensure detection validity.

Based on these observations, the number of bacterial colonies on the PSF membrane at 0.0 wt.% SNFs was higher compared to the membranes blended with SNFs. Figure 10b–e verifies that the number of bacterial colonies decreased with the increase in SNF concentration. Quantitative observation indicated that the numbers of bacterial colonies appearing on the membrane surface were 497, 448, 377, 169 and 82 for membranes at SNF concentrations of 0, 0.1, 0.2, 0.3 and 0.4 wt.%, respectively. This result is in agreement with previous findings in which silver nanoparticles decrease the number of bacterial colonies. Silver can modify the metabolism of bacterial activity by altering their DNA so that it loses its replication ability; hence, proteins become inactivated and the bacteria enter the death phase [81,82]. In addition, a significant reduction in the colonies formed was observed at 0.4 wt.% SNFs in the membrane. This can be explained by the higher SNF concentration leading to more surface contact between the silver and the bacteria. These findings conform to the statement that exposing bacteria to a higher concentration of nanoparticles will lead to a more dramatic decrease in the extent of the colonies’ growth [83,84].

## 4. Conclusions

In this study, we provided laboratory evidence for the efficacy of SNF-blended PSF membranes for water filtration purposes. We demonstrated that the introduction of SNFs changed the membrane surface hydrophilicity, pure water flux, protein rejection and antibacterial activity, resulting in PSF membranes with improved properties. The blended membrane with 0.1 wt.% SNFs exhibited low contact angles (55°) and the lowest antibacterial capability against *Escherichia coli*, as well as higher pure water flux and rejection against egg albumin. The best antibacterial property was related to the 0.4 wt.% SNF membrane. The blended PSF membranes developed in this study have a potential use in the water filtration process. Challenges regarding the optimal SNF concentration at their full scale and long-term usage remain and should be addressed in a pilot study.

## Figures and Tables

**Figure 1 polymers-14-03600-f001:**
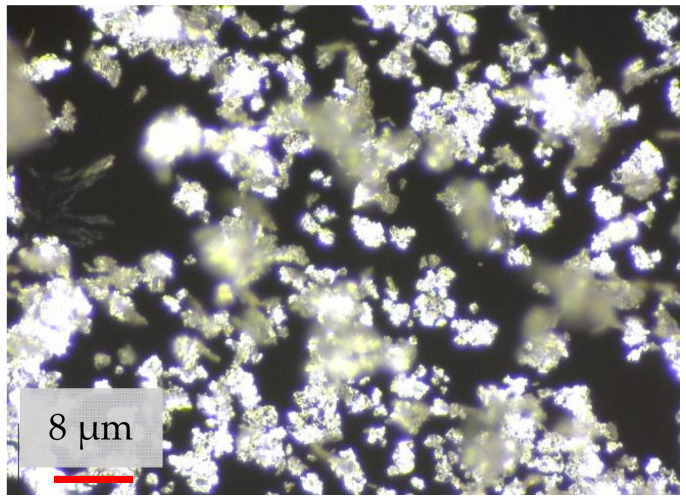
Image of silver nanoflake powder.

**Figure 2 polymers-14-03600-f002:**
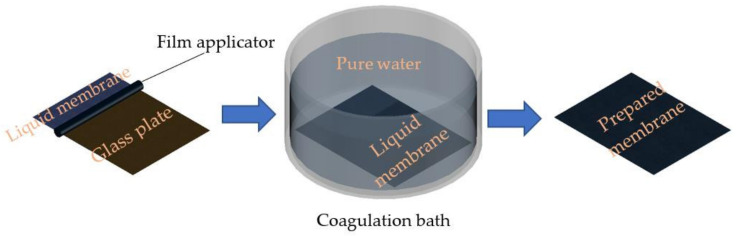
Membrane gelatinization process.

**Figure 3 polymers-14-03600-f003:**
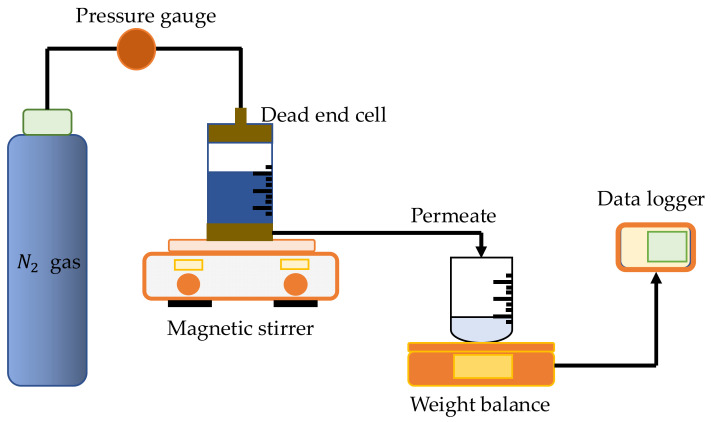
Experimental setup of the water flux test.

**Figure 4 polymers-14-03600-f004:**
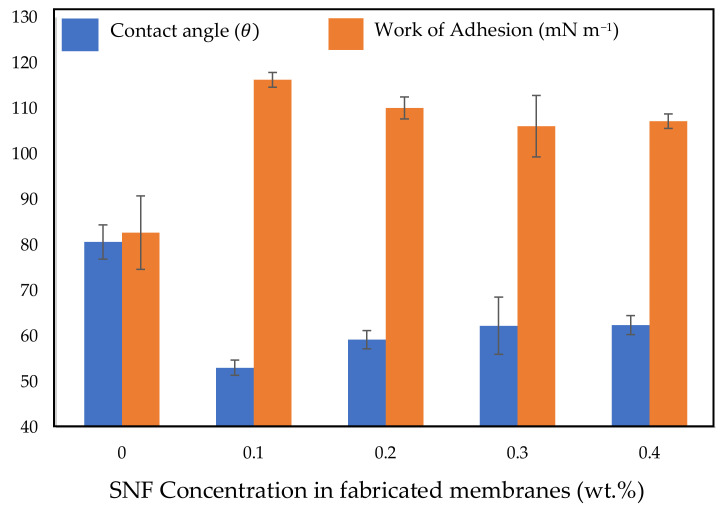
Contact angle and work of adhesion of the fabricated membranes at different SNF concentrations.

**Figure 5 polymers-14-03600-f005:**
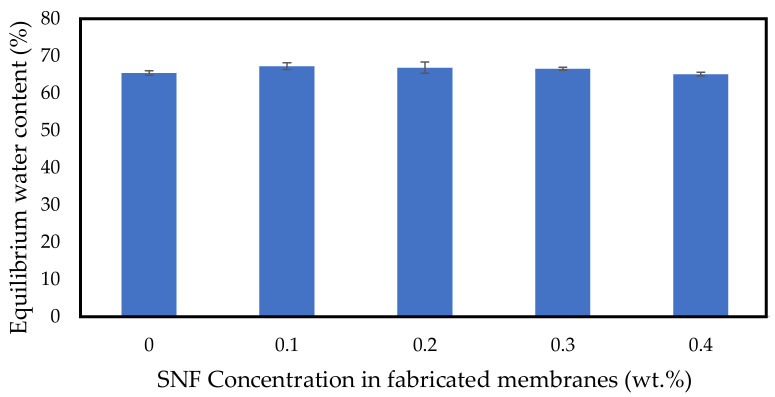
Equilibrium water content of the fabricated membranes at different SNF concentrations.

**Figure 6 polymers-14-03600-f006:**
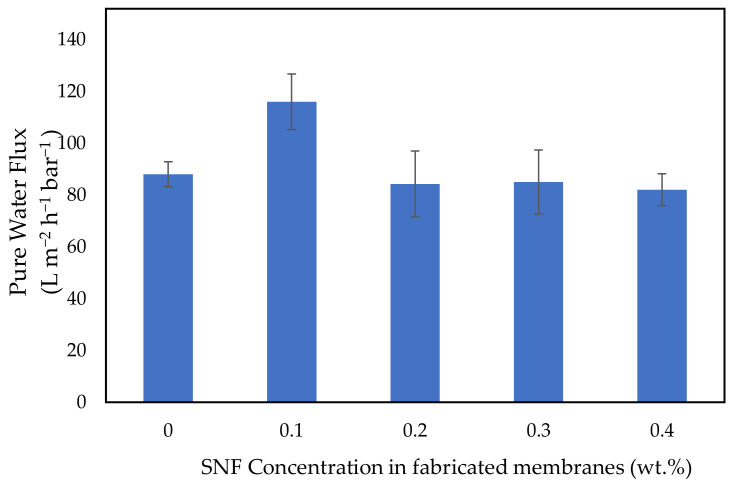
Pure water flux of the fabricated membranes at different SNF concentrations.

**Figure 7 polymers-14-03600-f007:**
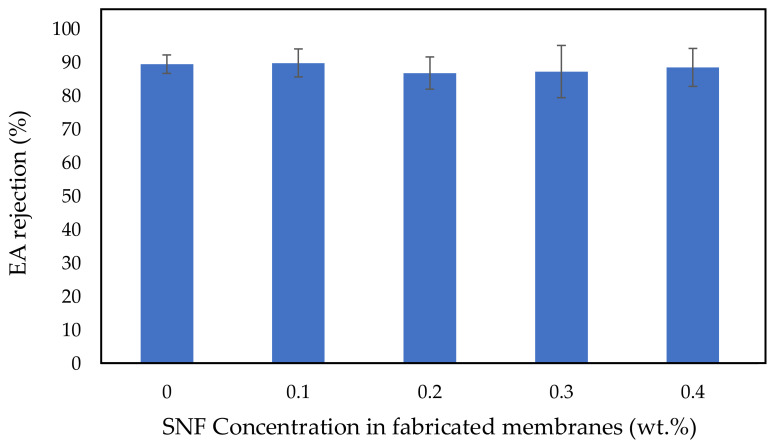
Effect of SNF concentration on EA rejection.

**Figure 8 polymers-14-03600-f008:**
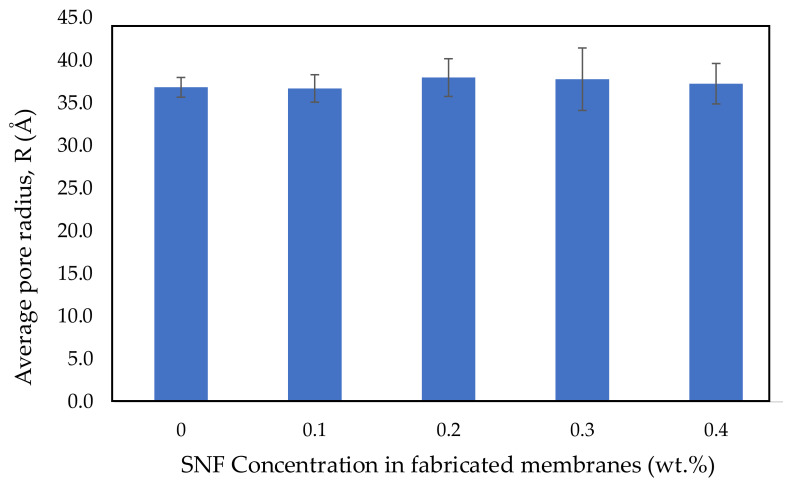
Effect of SNF concentration on the membrane average pore size.

**Figure 9 polymers-14-03600-f009:**
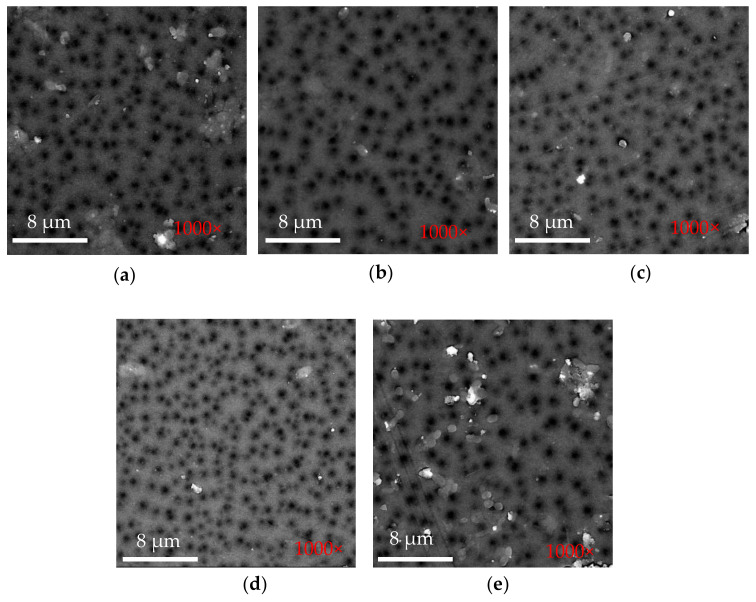
Surface SEM micrographs of (**a**) bare PSF (**b**) PSF-SNF 0.1 wt.%, (**c**) PSF-SNF 0.2 wt.%, (**d**) PSF-SNF 0.3 wt.% and (**e**) PSF-SNF 0.4 wt.% membranes.

**Figure 10 polymers-14-03600-f010:**
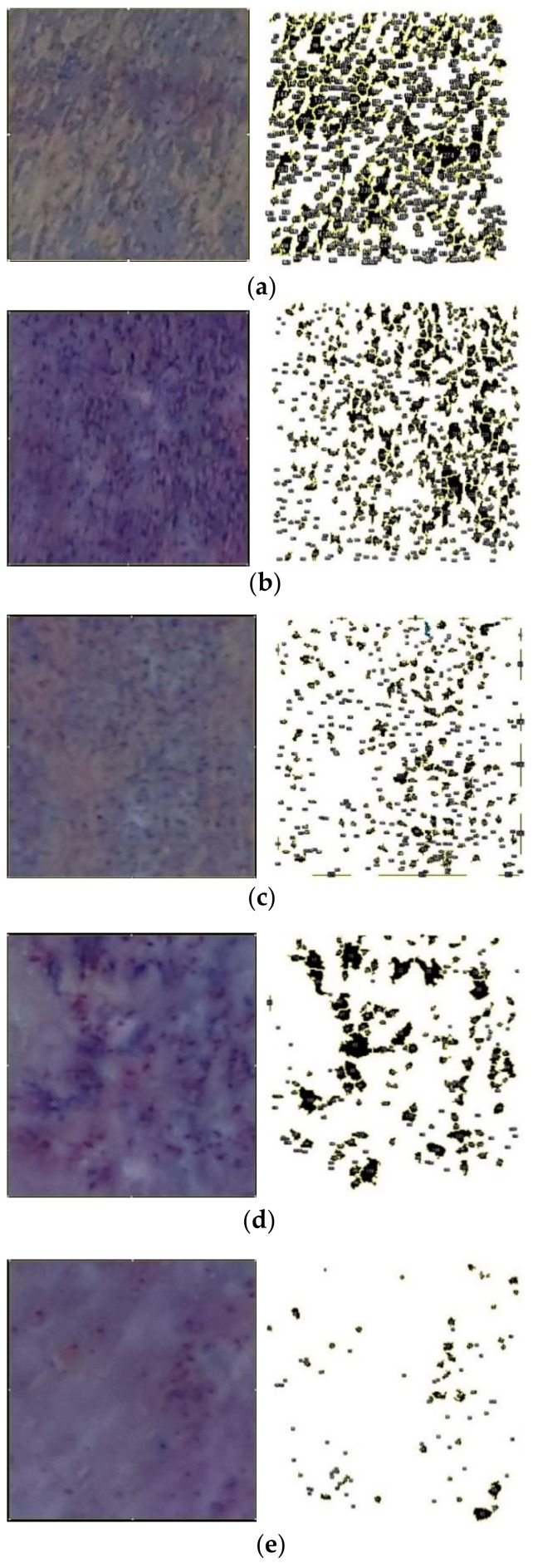
Results of the bacterial test on the fabricated membrane at different SNF concentrations: (**a**) 0 wt.%, (**b**) 0.1 wt.%, (**c**) 0.2 wt.%, (**d**) 0.3 wt.% and (**e**) 0.4 wt.%.

**Table 1 polymers-14-03600-t001:** Composition of the casting solution of PSF/SNF membranes.

Membrane	Blend Composition, wt.%		
	PSF	SNF	NMP
SNF 0	22	0	78
SNF 0.1	22	0.1	77.9
SNF 0.2	22	0.2	77.8
SNF 0.3	22	0.3	77.7
SNF 0.4	22	0.4	77.6

**Table 2 polymers-14-03600-t002:** Properties of sample water (river water).

pH	TDS (ppm)	Salinity (%)	Electrical Conductivity (µS/cm)	Temperature (°C)
8.5	143	0.01	293	28.1

**Table 3 polymers-14-03600-t003:** Performance of PSF/SNF membranes.

Membrane Code	Water Content (%)	Average Pore Radius, R (Å)	Egg Albumin Rejection (%)
SNF 0	65.4±0.6	36.9±1.2	84.2±2.6
SNF 0.1	67.2±0.9	36.7±1.6	87.4±3.3
SNF 0.2	66.8±1.5	38.1±2.2	73.3±7.7
SNF 0.3	66.5±0.4	37.8±3.6	78.9±12.1
SNF 0.4	65.1±0.5	37.3±2.3	79.7±10.8

**Table 4 polymers-14-03600-t004:** ANOVA statistical analysis of membrane equilibrium water content.

Equilibrium Water Content	Sum of Squares	Degree of Freedom	Mean Square	F Values	Significance
Between Groups	10.533	4	2.633	4.581	0.023
Within Groups	5.748	10	0.575		
Total	16.281	14			

## Data Availability

The data presented in this study are available from the corresponding author upon request.

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
