# Peer review of "Antibacterial Activity of Silver Nanoflake (SNF)-Blended Polysulfone Ultrafiltration Membrane"

_polymers, 2022, doi:10.3390/polym14173600_

Round 1

Reviewer 1 Report (Previous Reviewer 3)

The authors have addressed some of the comments of the reviewers and revised the manuscript accordingly. However, there are still some critical issues that need to be revised and the manuscript is not ready for publication.

1.      It is not recommended the use the SEM micrograph of AgNF obtained from the manufacturer company.  Authors should perform related characterizations. Therefore, Figure 1 should be replaced with the SEM analysis result and the Ref [63] should be removed from the manuscript.

2.      Table 1 should be moved to the experimental section.

3.      Although the authors stated that they made statistical analysis in the revised manuscript, it is seen that there was no such analysis in the revised version of the manuscript. Additionally, it should also have been described in the experimental section.   

4.      Without any statistical analysis it may be not correct to compare the numeral results obtained in the study. For example, there is no significant difference in water content values of the PS membranes containing different concentrations of AgNF and any correlation between the water angle and the Ag NF concentration.  However, In Section 3.2. it is written that “Bare PSF membrane has the maximum water content of 65.4± 0.9%, The water content of the membrane is slightly increased with the increase of SNF concentration. At SNF concentration of 0.1 wt.%, the membrane achieved the highest water content (67.2 ± 0.6%) as presented in Table 3, and became the most hydrophilic membrane compared to others. When the membrane hydrophilicity increases, the membrane is capable to transport more water through, hence increasing the water content of the membrane [69,78,79].

5.      Although some discussion was included (only in section 3.6, some few sentences) in the revised manuscript it is not sufficient. Results should be compared with the findings of the previously published articles on Ag NF or other metal nanoparticles-containing PS membranes.

Author Response

Reviewer 2 Report (Previous Reviewer 2)

Review of polymers-1862791

The authors have improved the manuscript significantly in terms of content and writing. This manuscript can be considered for publication, after addressing these issues:

  1. Line 214: Please add the details of the liquid media (ingredients, composition)
  2. Line 215: …Petri dish… -->  with uppercase P, without dash.
  3. Line 223-224, Table 2, row 1, column 4: The unit for electrical conductivity is µS/cm --> not uS/cm, but with µ (micro).
  4. Line 268: Please change LMHBar-1 as L m-2 h-1 bar-1 --> with lowercase m, h, b, respectively, and superscripted numbers for multiple units in the denominator.
  5. Figure 6, y axis: Please change LMHBar-1 as L m-2 h-1 bar-1 --> with lowercase m, h, b, respectively, and superscripted numbers for multiple units in the denominator.
  6. Reference 10: Please delete “Article number”
  7. Reference 14: Please write Fe3O4 with subscripted 3 and 4, not with subscripted F, e, and O.
  8. Reference 35:
  9. Reference 36: …addition… --> with lowercase a
  10. Reference 37: Please add a dot after the title and before the journal name.
  11. Reference 38: After the title and before the journal name, change the comma with a dot.
  12. Reference 39: Please add a dot after the title and before the journal name.
  13. Reference 64: Please write H2O with subscripted 2.

Round 2

Reviewer 1 Report (Previous Reviewer 3)

In the previous review report, it has been written that “It is not recommended to use the SEM micrograph of AgNF obtained from the manufacturer company.  Authors should perform related characterizations. Therefore, Figure 1 should be replaced with the SEM analysis result and the Ref [63] should be removed from the manuscript.

Although in the author's response letter it is written that “We have removed SEM image of AgNF from the manufacturer and replaced it with our own image of AgNF” in the revised manuscript the SEM image (Figure 1) was taken from another source was used. Interestingly, there is no such SEM image or Ag NF-related information in the cited reference.

Statistical analysis results are still not shown on the graphs. It is crucial to show by a symbol (*, **, ***)  if there is a significant difference with control group samples. 

Round 3

Reviewer 1 Report (Previous Reviewer 3)

The manuscript can be accepted in its present form.

This manuscript is a resubmission of an earlier submission. The following is a list of the peer review reports and author responses from that submission.

Round 1

Reviewer 1 Report

Although I do not see any serious queries from the manuscript, I cant see any novelty or anything new in this research. A simple google scholar search using 'Silver and polysulfone membranes' brings numerous output. It is no news that silver is anti-bacterial agent neither its use with PSF. I request the authors to demonstrate the uniqueness of this research.

Reviewer 2 Report

Review of polymers-1728661

This manuscript falls to the category of routine work, with low novelty, not-so-high-quality insights, incomplete characterizations. It can not be accepted for publication in Polymers.

  1. This manuscript must be proofread by native English speaker.
  2. Please write the units in X Y-1 format, instead of X/Y format.
  3. What is the size of the membrane used in this study (for the dead-end cell)? Is it 5 cm diameter? (or 19.625 cm2, assumed to be simplified to 20 cm2??) Please add this data.

  1. Section 2.6 and 3.5: The membranes’ MWCO must be further investigated by using several chemicals with various molecular weights, such as polyethylene glycol (PEG)-600, PEG-2000, PEG-5000, PEG-10000, PEG-20000 etc. Check Ling and Chung, Desalination 278 (2011) 194-202 https://doi.org/10.1016/j.desal.2011.05.019.

  1. Please change all “nAgF” to be “SNF” because nAgF might cause misleading as “nano silver fluoride”. SNF (or SNFs) really fits to be the abbreviation “silver nanoflakes”.

-Location: Line 2, 16, 17, 18, 21, 22, 25, 26, 26 again, 27, 28, 29, 103, 104, 106, 126, 128, 182, 202, 204, 205, 206, 206 again, 207, 209, 210, 213, 217, 218, 220, 223, 226, 226 again, 227, 228, 232, 235, 236, 237, 237 again, 238, 243, 247, 249, 250, 253, 257, 262, 267, 267 again, 268, 270, 271, 301, 306, 306 (again), 309, 312.

-Location: Figure 3, 4, 5, 6, 7, 8.

  1. Line 21: Escherichia coli --> scientific names must be written in italic, started uppercase letter for the genus, and lowercase letter for the species name.
  2. Line 22, 186, 191: Please corroborate about ”river water”, where was it taken from (location, city, province), the characteristics of the river water (COD, BOD, TSS, DO, TOC, color, microbiological colony forming unit per mL (CFU mL-1), etc.
  3. Line 23: …55°, that is less than that of the pristine PSF membrane (81°), exhibiting the…
  4. Line 25: ..enlarging the..
  5. Line 27: E. coli --> scientific names must be written in italic, started uppercase letter for the genus, and lowercase letter for the species name --> “coli” with lowercase c, not uppercase C.
  6. Line 27: …E. coli, where the killing rate was depending…
  7. Line 30: … utilizeD..
  8. Line 43: superior chemical resistance --> REALLY? Polymeric membrane will be destroyed in HCl 1 M, while ceramic membrane will not be.
  9. Line 63-66 must be revised heavily. It was claimed that modification by polymer blending (i.e. involving TWO polymers or more) is advantageous. However, this manuscript does not perform polymer blending. It is more suitable to add another entry about addition of additives.
  10. Line 63-66: Reference 46 must be written besides reference 40 and 41, in the line 63 (and then the reference number must be revised).
  11. Line 63-66: The revision has to be like this: …such as polymer blending [Ref 40, 41, 46], incorporating nanoparticles [Ref], surface chemical modification [Ref], interfacial polymerization, and addition of additives. It is known that the additives...

--> Delete Line 64-66 about polymer blending.

  1. Line 63-66: For interfacial polymerization on PSF:

Chem. Eng. Sci. 80 (2012) 219-231 https://doi.org/10.1016/j.ces.2012.05.033

IOP Conf. Ser. Earth Enviro. Sci. 109 (2017) 012042 https://doi.org/10.1088/1755-1315/109/1/012042

  1. Line 76: ..shapes…
  2. Line 82: …properties…
  3. Line 85-87 and Line 89-92 must be merged together in order to avoid the formation of short paragraph(s).
  4. Line 107: If the PSF concentration is 22%, and additives <0.4%, then what would be the remaining 77-78%? Is it NMP? Please make a table of the composition of the polymeric dope solution.

  1. Line 111: …E. coli…--> scientific names must be written in italic, started uppercase letter for the genus, and lowercase letter for the species name --> “coli” with lowercase c, not uppercase C.
  2. Line 116: …N-Methyl-2…
  3. Line 129: …200 µm…
  4. Line 136: AutoCAD software?? This is a software for civil engineers, mechanical engineers, architects. So strange if it is employed to measure contact angle. Do you mean ImageJ?
  5. Line 139: …work of adhesion…
  6. Equation 1, 2, 3 and 4: Please write the equation using Microsoft Equation, and with uniform font type and font size.
  7. Line 164: Please add the reason(s) and reference(s) about the use of 280 nm wavelength, and not other number.
  8. Line 185: Gram-negative --> uppercase G, because it is originated from the name of a person, a distinguished microbiologist Hans Christian Gram.
  9. Line 186: …E. coli…--> scientific names must be written in italic, started uppercase letter for the genus, and lowercase letter for the species name --> “coli” with lowercase c, not uppercase C.
  10. Line 186: How about the effect of Gram-positive bacteria the membrane performance? Please add the data. You can use Staphylococcus aureus.
  11. Line 186 and Line 191: Please corroborate about ”river water”, where was it taken from (location, city, province), the characteristics of the river water (COD, BOD, TSS, DO, TOC, color, microbiological colony forming unit per mL (CFU mL-1), etc.
  12. Section 2 and 3: Add SEM characterization, especially for the cross sectional view, as the morphology characterization of the membranes.
  13. Section 2 and 3: Add FTIR, EDX, XPS wide scan, XPS narrow scan characterizations as the chemical characterization of the membranes.
  14. Figure 3: Change “Adhesion work” to be “Work of adhesion”. Unit: 10-3 N m-1 --> in order to avoid confusion of “m divides m”.
  15. Section 3.2: Perform statistical analysis.
  16. Line 217: …concentration of SNF. It shows that…of the membranES is not significantly changing, due to small amount of SNF added (0.1-0.4% only). Nevertheless, the membrane surfaces experience considerable changes in hydrophilicity.
  17. Line 227: Why 0.1 wt% SNF provides the highest PWP, almost 120 L m-2 h-1 bar-1? Please provide reason(s) and reference(s).
  18. Line 235-241 must be revised. Add statistical analysis. Suggestion for the revision: …in Figure 6. It can be observed that the EA rejection is not significantly changed at various addition of SNF, since the addition of SNF is less than 0.5 wt%. --> delete line 239-241.
  19. Line 245: …summary of the average pore size of the membranes.
  20. Delete line 250-251.
  21. Line 310: …E. coli…--> scientific names must be written in italic, started uppercase letter for the genus, and lowercase letter for the species name --> “coli” with lowercase c, not uppercase C.

  1. Please write reference with uniform style. There are some journal names that are written in italic, some not in italic. The journal name in the reference 34 has “Environ.” not in italic, but “Nanotechnol. Monit. Manag.” in italic.
  2. No need to write “Article number”.
  3. Reference 13: Please check and revise.
  4. Reference 15: BaTiO3 --> subscripted 3
  5. Reference 22 vs 23: Two references from the same authors, but with different style of writing the name of the authors.
  6. Reference 28: MoS2 --> subscripted 2
  7. Reference 29: SiO2 --> subscripted 2
  8. Reference 40: Escherichia coli --> scientific names must be written in italic, started uppercase letter for the genus, and lowercase letter for the species name.
  9. Reference 49: TiO2 --> subscripted 2
  10. Reference 52: Fashandi1? Delete “1”.
  11. Reference 57: Escherichia coli --> scientific names must be written in italic, started uppercase letter for the genus, and lowercase letter for the species name.
  12. Reference 57: Gram-negative --> uppercase G, because it is originated from the name of a person, a distinguished microbiologist Hans Christian Gram.
  13. Many references are outdated, not from recent five years.

Reviewer 3 Report

The article is on the preparation and characterization of the silver nanoflake-containing polysulfone ultrafiltration membranes. The language of the study should be improved; grammatical errors should be corrected by a native speaker. Specific comments are listed below:

  1. Introduction: The sentence “Silver is known for its good conductivity, chemical stability, and antibacterial activity” should be cited. The term conductivity should be revised as electrical and thermal conductivity.
  2. Introduction: “Moreover, the membranes incorporated with nAg showed membranes with different porosities and nanoparticle sizes” Sentence should be corrected. “………..with nAgs showed different porosities as a function of additive particle size”
  3. Introduction: The novelty of the study should be described in detail. There are some other studies [54-56] in the literature on the preparation of AgNP-containing PS membranes. Authors should discuss the advantage of using Ag NF instead of AgNP.
  4. Experimental: The membrane fabrication section should be written in detail. “Solvent utilized, PS concentration, etc should be written. The term “electric lab mixer” should be revised (magnetic stirrer?) The details and conditions of the gelatinization process should be given.
  5. Results- The position of the labels in Fig.3 should be adjusted.
  6. Results- The reason for the increase in contact angle of the nAgF-PSF membrane with the increase of nAgF concentration should be explained and discussed.
  7. Results- Fig 4 “The figure shows the water content of the membrane is slightly increased with the increase of nAgF concentration”. The mentioned increase in water content cannot be detected from the graph. Numerical values should be given in the text. Additionally, statistical analysis is necessary for this statement.
  8. Results- The same comment written for Fig.4 is also valid for Figure 6 and Figure 7. They should be discussed in detail and statistical analysis should be performed for a concluding statement.     
  9. Results- Results shown in Figure 7 need explanation. Error bars should be included. The difference in the pore as a function of AgNF concentration should be discussed.
  10. There is no discussion in the manuscript. Results should be compared with the findings of the previously published articles on Ag or other metal nanoparticles-containing PS membranes.
  11. Conclusions- It is written that “The blended membrane (0.1 wt.% of nAgF) exhibited low contact angles (55°), promising antibacterial properties against E. Coli…” However, the antibacterial property observed at 0.1 AgNF is too low (Almost similar to AgNF-free membranes).